# Analysis of the impact of different landing heights and lateralities on lower limb joint load based on statistical parametric mapping

Lian Duan[1,2☉], Yuan Gao [iD][2]*, Feng Gao[1]*, Yang Sun[2]*, Naiyuan Tian[2‡]

1 China Volleyball College, Beijing Sport University, Beijing, China, 2 School of Physical Education, Yanshan University, Qinhuangdao, Hebei, China

☉ These authors contributed equally to this work.
‡ NT also contributed equally to this work.
* gaoyuan197812@126.com (YG); gaofeng202502@126.com (FG); syang0217@126.com (YS)

## Abstract

This study aimed to investigate the effects of landing height and limb laterality on lower limb joint loading using Statistical Parametric Mapping (SPM), thereby providing insights for injury prevention and optimizing performance in dynamic sports. Twenty male participants were assessed using the Qualisys 3D motion capture system and Kistler 3D force plates during landings from 30 cm and 45 cm on both dominant and non-dominant legs. No significant interaction effects between landing side and height were observed, but landing on the non-dominant leg increased the ankle joint internal rotation moment. The knee's abduction moment significantly increased during the 17%−37% landing phase, and the hip's abduction moment significantly increased during the 86%−100% phase. Higher landing height significantly increased vertical ground reaction force (GRF) during the mid-buffering phase (23%−35%). Landing on the dominant side was linked to an elevated risk of lower limb joint injuries due to increased abduction moments at the hip, external rotation moments at the knee, and internal rotation moments at the ankle. Furthermore, greater landing height correlated with higher vertical GRF, intensifying stress on the lower limbs and increasing injury risk. To reduce the risk of injury, training should focus on increasing strength and control of the non-dominant leg and controlling landing height.

## Introduction

In recent years, rope skipping has emerged as a widely adopted exercise modality, owing to its low cost, ease of learning, and cardiovascular benefits. Its effectiveness in improving both aerobic fitness and coordination has made it a favored exercise among fitness enthusiasts and recreational athletes alike [1]. Freestyle rope skipping, known for its diverse techniques and visually engaging performances, has gained widespread popularity. However, the complexity of these movements elevates the

**Data availability statement:** All relevant data are within the manuscript and its Supporting information files.

**Funding:** The author(s) received no specific funding for this work.

**Competing interests:** The authors have declared that no competing interests exist.

risk of injury, particularly during the landing phase. Recent studies highlight a notably high incidence of lower limb injuries (87.5%), with knee and ankle joints being most frequently affected, comprising approximately 80% of all injury cases [2–4]. Despite the human body's inherent structural symmetry, functional asymmetry, or laterality, is well-documented in lower limb biomechanics [5,6]. Excessive lateral laterality has been identified as a potential risk factor for non-contact joint injuries [7,8]. However, most rope skipping research assumes limb symmetry, primarily analyzing data from a single limb to assess landing-related injury risks [9–12]. Consequently, there is limited understanding of the biomechanical differences between dominant and non-dominant limbs during single-leg landing in rope skipping, highlighting a critical gap in the literature.

Statistical Parametric Mapping (SPM), based on random field theory, utilizes topological analysis to examine dynamic changes and interrelationships in time series data, enhancing the robustness of biomechanical conclusions, as indicated by Wattananon et al. [13] and Pataky et al. [14–16]. While traditional zero-dimensional statistical methods can identify feature differences, they often fail to capture dynamic patterns in time series data, as highlighted by previous studies [16–18]. To address these limitations, this study aimed to systematically analyze the effects of landing height and limb lateralities on lower limb joint loading using Statistical Parametric Mapping (SPM). Specifically, the study focused on biomechanical differences between the dominant and non-dominant limbs during landings from different heights and their subsequent impact on joint loading. By addressing this gap in the literature, the study aimed to provide scientific evidence for injury prevention in dynamic activities such as rope skipping and to optimize training strategies.

## Materials and methods

### Participants

Study participants were recruited between 15 April 2024 and 23 April 2024 through campus bulletin board postings and social media platforms targeting university students at Yanshan University, Qinhuangdao, Hebei Province. The sample size was determined using G*Power software with the following parameters: effect size f = 0.4, α error probability = 0.05, power (1-β) = 0.8, number of groups = 1, number of measurements = 2, and correlation among repeated measures = 0.5. The minimum required sample size was calculated to be 15 participants [19]. To account for potential data loss due to movement failure, marker displacement, or invalid data, a 30% redundancy was added, Thus, 20 male rope-skipping enthusiasts participated in this study (age: 19.75 ± 0.86 years, height: 181.24 ± 4.62 cm, body mass: 72.21 ± 7.66 kg, BMI: 22.06 ± 2.94 kg/m²). Data collection was completed during a 12-day experimental period following recruitment.

All participants were recruited based on the following inclusion criteria: (1) no congenital lower extremity deformities or acquired musculoskeletal disorders; (2) no history of lower extremity joint injuries within the past 6 months; and (3) sufficient physical ability to perform the task of stair walking. Exclusion criteria included (1) a history of lower extremity surgery; (2) use of medications affecting neuromuscular

function within the past 12 months; (3) the presence of neuromuscular, psychologic, or cardiovascular disorders; and (4) the presence of acute pain symptoms during the test. All procedures were in accordance with the Declaration of Helsinki. Participants were fully informed of the purpose and procedures of the study and written informed consent was obtained. The study protocol was approved by the Ethics Committee of Qinhuangdao First Hospital.

The determination of the dominant side was conducted through a kicking test, where participants were asked to kick with both feet. The foot that achieved the greater kicking distance was identified as the dominant side [7,20].

### Single-leg landing test

Participants were instructed to place their hands on their hips and stand 5 cm from the force platform, with jumping boxes positioned at heights of 30 cm and 45 cm. Upon receiving a "start" command, participants lifted their test leg and held it in the air. They then leaned forward and descended by inertia, landing on the test foot to maintain posture and stability [2,21]. Participants were required to focus visually on a target displayed on a blank wall in front of them throughout the test. Prior to the formal test, participants underwent a 6-minute familiarization process, including a movement demonstration and practice trials. Additionally, a 10-minute warm-up was conducted, consisting of 5 minutes of jogging and dynamic stretching, followed by 5 minutes of light jumping and landing exercises to prepare participants for the task. Trials were repeated if participants touched the ground with the non-test leg or used their arms for balance (Figs 1 and 2) [22].

Participants stood on the jumping box with one leg lifted, then leaned forward and landed on the test leg while keeping their hands on their hips. The two heights correspond to the experimental conditions used for data collection.

(A) The participant stands on the jumping box with the test leg suspended in the air and both hands placed on the hips, preparing to descend. (B) The test leg begins to descend, and the body leans forward, utilizing inertia to approach the ground. (C) The test leg makes contact with the ground, maintaining a stable posture, completing the test.

### Data collection

During trials, participants wore study-specific tights, and after warming up, 20 infrared reflective markers were placed on anatomical landmarks according to the CAST (Calibrated Anatomical Systems Technique) lower limb model, which employs static calibration and marker tracking to establish joint coordinate systems while minimizing soft tissue artifacts [23]. All markers were applied by the same experienced and professionally trained researcher, following standardized procedures and anatomical landmark definitions to ensure data accuracy and consistency. Following the collection of static model data, eight markers were removed from the medial and lateral sides of the knee and ankle joints. Participants performed at least three complete movements during trials, and kinematic and kinetic data were synchronously captured using the Qualisys 3D motion capture system (8 cameras, 200 Hz) and a Kistler 3D force platform (2000 Hz) during landing (Fig 3).

The test apparatus included: a Qualisys 3D motion capture system to capture motion in space (A), a Kistler 3D force table to measure changes in force (B), and an experimental test site (C).

### Data processing

In this study, the landing cushion phase (the moment of contact of the test leg on the ground up to the maximum angle of knee flexion of the supporting leg) of the drop jump movement was recorded. The raw data were pre-processed using the QTM software supplied with the Qualisys system, and hip, thigh, calf, and foot coordinate systems were established based on the CAST lower limb model, using infrared reflective markers to create a simulation of the subject's lower limb. Kinematic data were subsequently imported into Visual 3D software for further processing. A fourth-order Butterworth low-pass filter with a cutoff frequency of 12 Hz was applied to smooth the trajectory data, as described by Kristianslund et al. [24], Lenz et al. [25] and Monfort-Torres et al. [21]. Net internal joint moments for ankle plantarflexion, knee flexion,

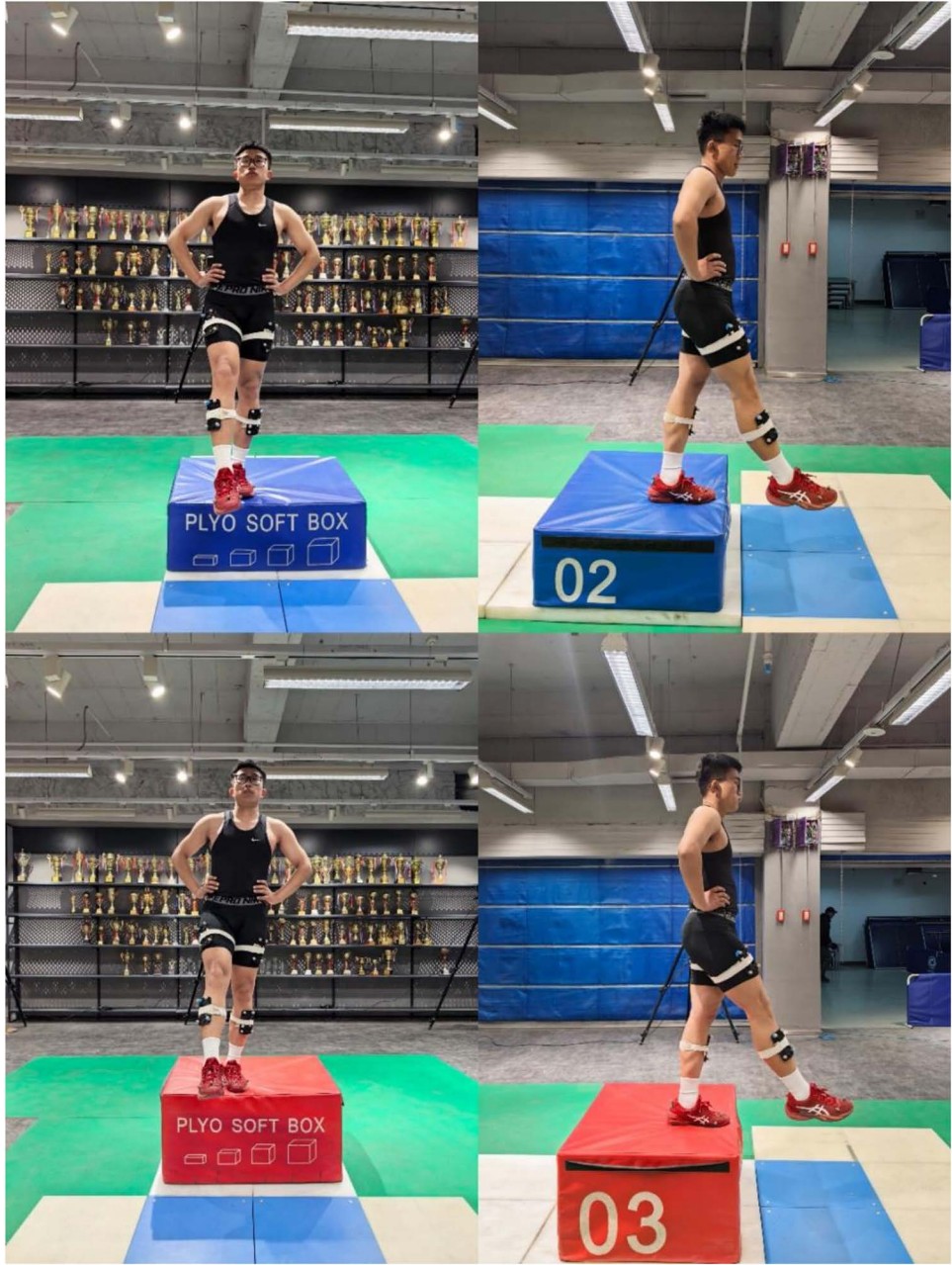

**Fig 1. Participants performing the single-leg landing task from two different heights (30 cm and 45 cm).**

and hip flexion (N•m•kg) were defined as positive (+) values, while ankle dorsiflexion, knee extension, and hip extension moments were defined as negative (−) values. Ground reaction forces (N) and joint moments (N•m•kg) Lenz et al. [25] were calculated through inverse dynamics (Fig 4).

The landing cushioning step is defined as the moment from the initial foot contact with the force platform to the maximum knee flexion angle, A: the moment from the initial foot contact with the force platform, B: the moment from the maximum knee flexion angle, C: the division of the landing cushioning phase.

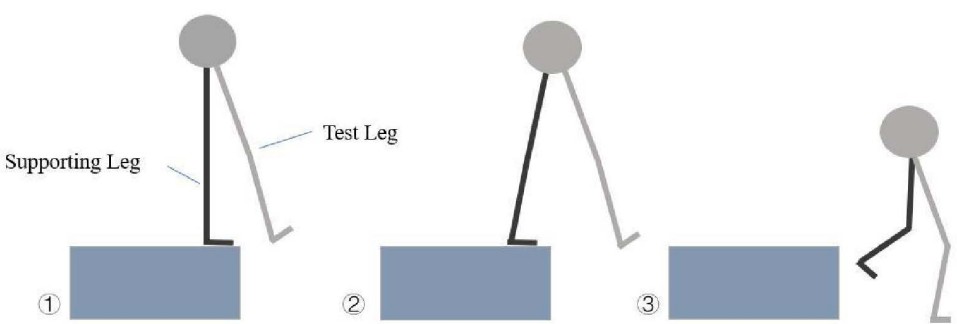

**Fig 2. Single leg drop test manoeuvre.**

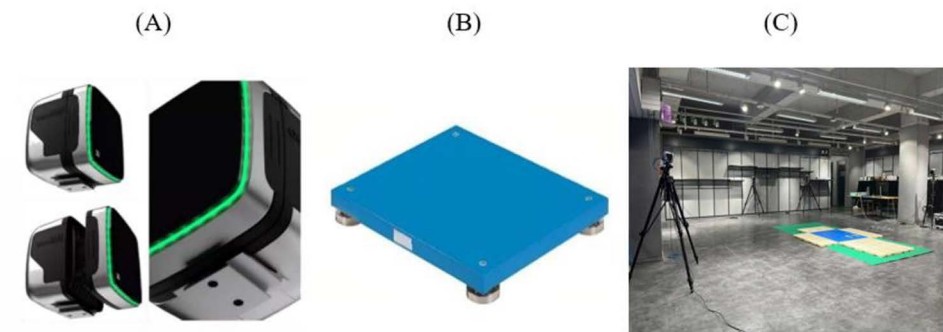

**Fig 3. Qualisys 3D Motion Capture System and Kistler 3D Force Stage.**

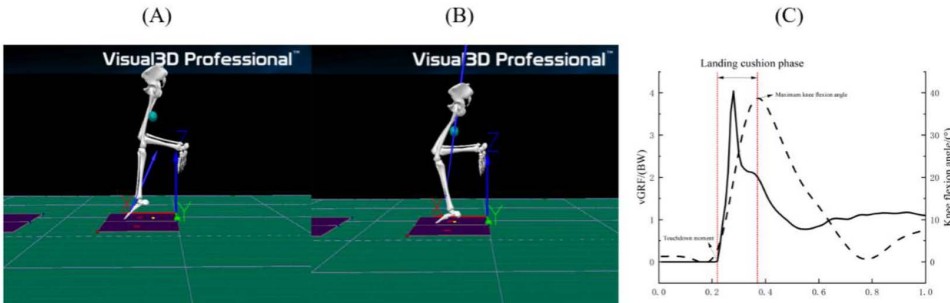

**Fig 4. Landed buffer phase.**

## Statistical analysis

This study analyzed biomechanical indicators related to lower limb joint loads during the landing cushioning phase, defined as the period from initial ground contact by the supporting leg to maximum knee flexion. Net internal joint moments were normalized to 101 data points across the landing phase for precise temporal resolution. Mean value curves were generated to compare dynamic variations in joint load related parameters. A two-way ANOVA was employed to examines the effects of side and landing height on peak vertical GRF and 3D net internal joint moments during the

cushioning phase, with normality and homogeneity of variance verified. SPM{F} analysis assessed side and landing height effects on vertical GRF and 3D joint moment time series. Statistical significance was set at $p < 0.05$. Analyses were performed with SPSS 23.0 and Origin 2022.

## Results

Two-way ANOVA and SPM{F} analyses were performed to examine the effects of laterality (dominant vs non-dominant) and landing height (30 cm vs 45 cm) on lower extremity joint loads. The results showed that there was no significant interaction between laterality and height ($p > 0.05$). This suggests that these two factors independently affect joint loads and have no combined effect.

### Ground reaction forces in the landing cushioning phase

The two-way ANOVA results indicated that the peak vertical GRF during the landing cushioning phase was significantly greater for the 45 cm drop height (4.32 BW) compared to the 30 cm drop height (3.48 BW), independent of the side factor, with statistical significance denoted by $p < 0.001$ and $F = 30.914$ (Table 1).

 The SPM{F} analysis revealed that during the 23% to 35% of the landing cushioning phase, the vertical GRF for the 45 cm drop height was significantly greater than that for the 30 cm drop height, regardless of the side factor, with $p = 0.004$ and $F = 8.927$ (Fig 5).

 Mean ensemble curves for vGRF (A) and their respective SPM{F} outputs (B, C, D) from 0 to 100% of stance phase. The dashed horizontal line on the SPM{F} outputs indicates the critical threshold boundary. The grey, shaded regions indicates the significant supra-threshold clusters representing statistically significant differences.

**Table 1. Comparison of vertical GRF peaks during the landing cushion phase for different side groups and drop heights (BW).**

|  | D30 | D45 |
|---|---|---|
| **Non-dominant** | 3.47 ± 0.48# | 4.31 ± 0.64 |
| **Dominant** | 3.49 ± 0.58 | 4.34 ± 0.67 |

Mean (Standard difference), *:There was a significant difference between D30 and D45.

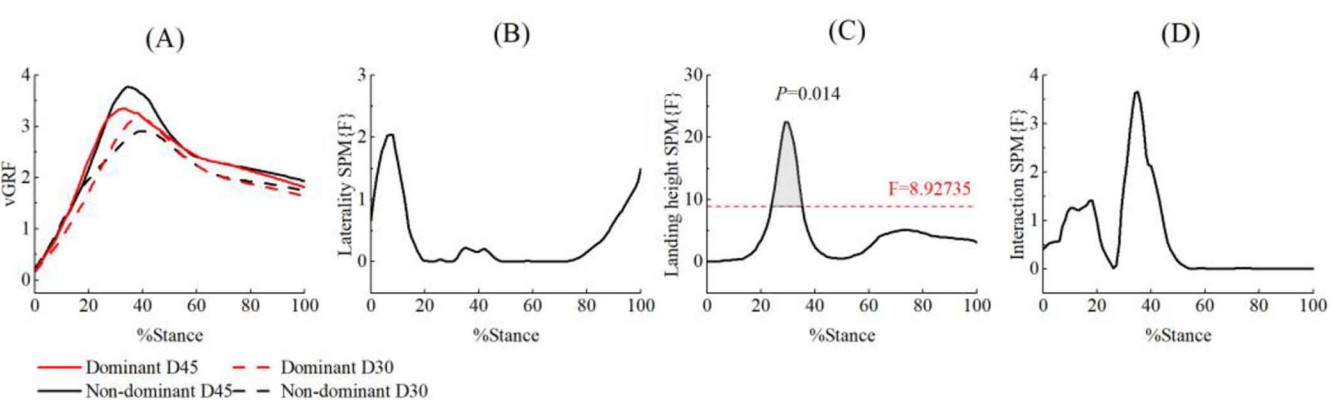

**Fig 5. Comparison of vertical GRF SPM{F} in the landing cushion phase for different side and drop heights.**

## Lower limb kinetic characteristics during the landing cushioning phase

The results of the two-way ANOVA indicated that during the landing cushioning phase, the peak dorsiflexion moment of the ankle joint (0.99 N•m•kg$^{-1}$ vs. 1.64 N•m•kg$^{-1}$, $p = 0.01$, F = 6.901), the peak internal rotation moment of the ankle joint (0.11 N• m•kg$^{-1}$ vs. 0.37 N•m•kg$^{-1}$, $p < 0.001$, F = 47.348), the peak abduction moment of the knee joint (0.36 N•m•kg$^{-1}$ vs. 0.88 N•m•kg$^{-1}$, $p = 0.006$, F = 7.834), and the peak external rotation moment of the knee joint (0.19 N•m•kg$^{-1}$ vs. −0.41 N•m•kg$^{-1}$, $p < 0.001$, F = 58.042), were significantly greater on the non-dominant side compared to the dominant side. In addition, irrespective of the side factor, the peak flexion moment of the knee joint when landing from a 45 cm height was significantly greater (2.66 N•m•kg$^{-1}$ vs. 2.03 N•m•kg$^{-1}$, $p = 0.003$, F = 9.132) than when landing from a 30 cm height (Table 2).

The SPM{F} analysis indicated that during the 86% to 100% of the landing cushioning phase, the peak hip abduction moment was significantly greater on the non-dominant side compared to the dominant side ((F = 8.084, $p = 0.022$), irrespective of the height factor (Fig 6). Similarly, during the 17% to 37% of the landing cushioning phase, the peak abduction moment of the knee joint was significantly greater on the non-dominant side ((F = 7.167, $p = 0.026$), independent of height factor (Fig 7). Throughout the entire landing cushioning phase, the ankle internal rotation moment on the non-dominant side was significantly greater than on the dominant side (F = 8.127, $p < 0.001$) (Fig 8). In addition, during the 10% to 38% of the landing cushioning phase, the knee joint flexion moment was significantly greater for a 45 cm landing (F = 8.350, $p < 0.001$) than for a 30 cm landing, regardless of the side factor (Fig 7).

Mean ensemble curves for 3D hip joint moment (A) and their respective SPM{F} outputs (B, C, D) from 0 to 100% of stance phase. The dashed horizontal line on the SPM{F} outputs indicates the critical threshold boundary. The grey, shaded regions indicates the significant supra-threshold clusters representing statistically significant differences. Positive sagittal, frontal, and transverse plane knee joint kinetics represent internally applied knee extension/adduction/internal rotation moments.

Mean ensemble curves for 3D knee joint moment (A) and their respective SPM{F} outputs (B, C, D) from 0 to 100% of stance phase. The dashed horizontal line on the SPM{F} outputs indicates the critical threshold boundary. The grey, shaded regions indicates the significant supra-threshold clusters representing statistically significant differences. Positive sagittal, frontal, and transverse plane knee joint kinetics represent internally applied knee extension/adduction/internal rotation moments.

Mean ensemble curves for 3D ankle joint moment (A) and their respective SPM{F} outputs (B, C, D) from 0 to 100% of stance phase. The dashed horizontal line on the SPM{F} outputs indicates the critical threshold boundary. The grey,

Table 2. Comparison of the peak moments of the lower limb joints during the landing cushioning phase by side and height of fall.

| Peak moment (N·m·kg$^{-1}$) | Dominant | | Non-dominant | | Interaction effect $p$ value | Main effect of laterality $p$ value | Main Effect of Height $p$ value |
|---|---|---|---|---|---|---|---|
| | D30 | D45 | D30 | D45 | | | |
| Ankle dorsiflexion | 0.95 ± 1.39* | 1.04 ± 1.51 | 1.65 ± 0.52 | 1.62 ± 0.49 | 0.797 | 0.010 | 0.916 |
| Ankle valgus | 0.46 ± 0.36 | 0.60 ± 0.45 | 0.49 ± 0.41 | 0.52 ± 0.34 | 0.538 | 0.849 | 0.337 |
| Ankle internal rotation | 0.11 ± 0.15* | 0.11 ± 0.18 | 0.36 ± 0.17 | 0.38 ± 0.17 | 0.862 | 0.000 | 0.878 |
| Knee flexion | 1.90 ± 1.12# | 2.42 ± 1.15 | 2.16 ± 0.65 | 2.89 ± 0.72 | 0.598 | 0.077 | 0.003 |
| Knee valgus | 0.20 ± 0.71* | 0.23 ± 0.98 | 0.90 ± 0.56 | 0.85 ± 0.96 | 0.549 | 0.006 | 0.391 |
| Knee internal rotation | 0.22 ± 0.33* | 0.17 ± 0.47 | −0.30 ± 0.22 | −0.51 ± 0.35 | 0.327 | 0.000 | 0.107 |
| Hip flexion | 0.52 ± 2.21 | 0.74 ± 3.17 | 0.57 ± 1.72 | 2.05 ± 2.14 | 0.239 | 0.204 | 0.117 |
| Hip valgus | 1.79 ± 1.47 | 1.80 ± 1.44 | 1.93 ± 0.49 | 2.13 ± 0.47 | 0.707 | 0.345 | 0.672 |
| Hip internal rotation | 0.61 ± 0.31 | 0.64 ± 0.57 | 0.78 ± 0.30 | 0.83 ± 0.46 | 0.968 | 0.065 | 0.658 |

Joint Moment (Peak Joint Moment), Mean (Standard difference), *: Significant difference between dominant and non-dominant side, #: There was a significant difference between D30 and D45.

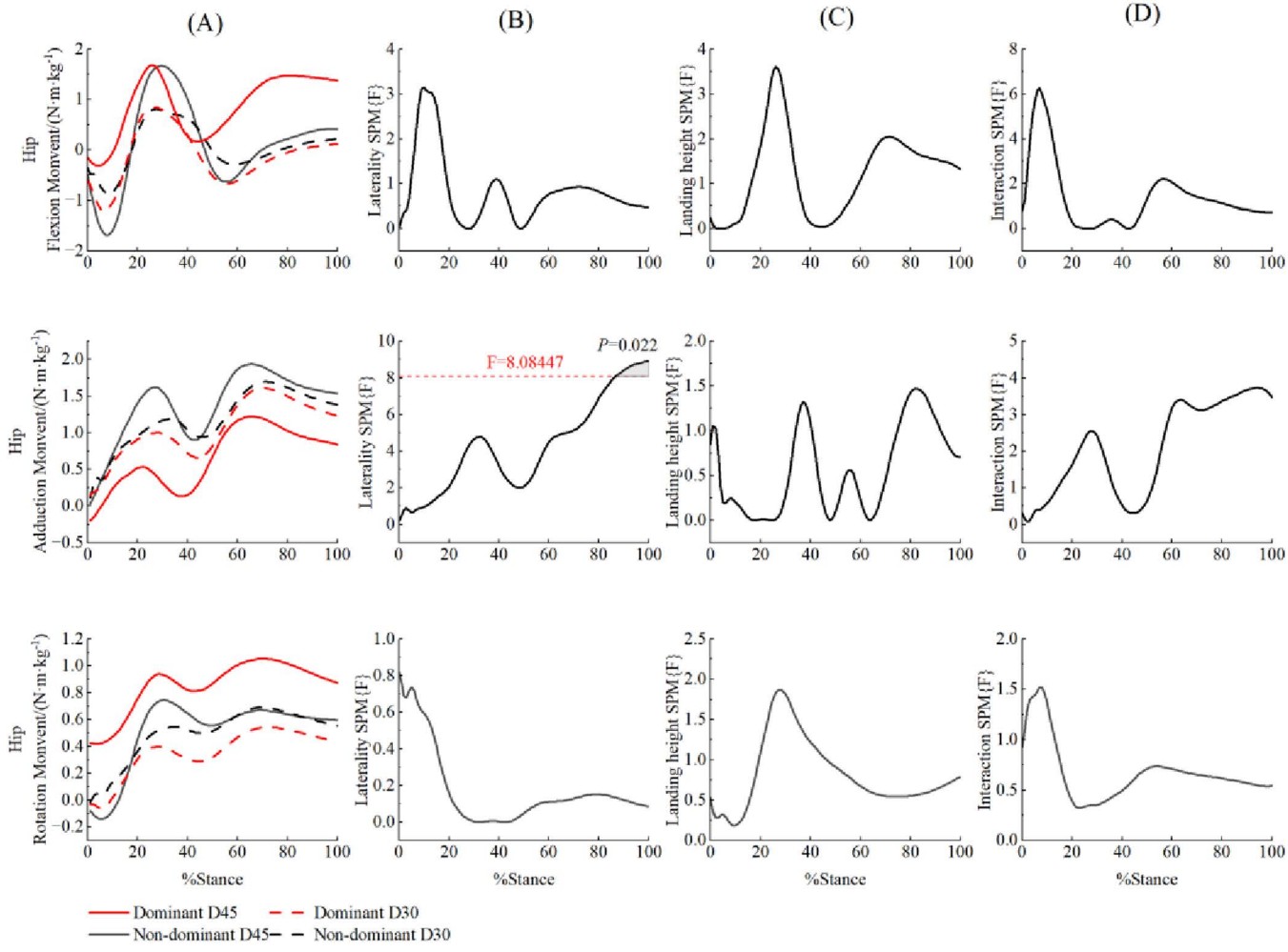

**Fig 6. Comparison of hip moments SPM{F} during the landing cushion phase for different side differences and drop heights.**

shaded regions indicates the significant supra-threshold clusters representing statistically significant differences. Positive sagittal, frontal, and transverse plane knee joint kinetics represent internally applied knee extension/adduction/internal rotation moments.

## Discussion and analysis

Previous studies have primarily focused on single-limb mechanics, assuming bilateral symmetry, which limits understanding of limb asymmetries in dynamic activities. Given the high incidence of lower limb injuries, examining these differences is crucial. This study aimed to investigate the effects of landing height and limb lateralities on joint loading, addressing this gap and providing insights for injury prevention and performance optimization.

### Comparative analysis of side differences during the landing cushioning phase

Two-way ANOVA showed no significant difference in peak 3D hip moments regarding side or height, but SPM{F} revealed higher hip abduction moments on the non-dominant side, reflecting the need for stability during maximum knee flexion and impact absorption [20]. During landing, increased hip abduction moment contributes to maintaining postural stability

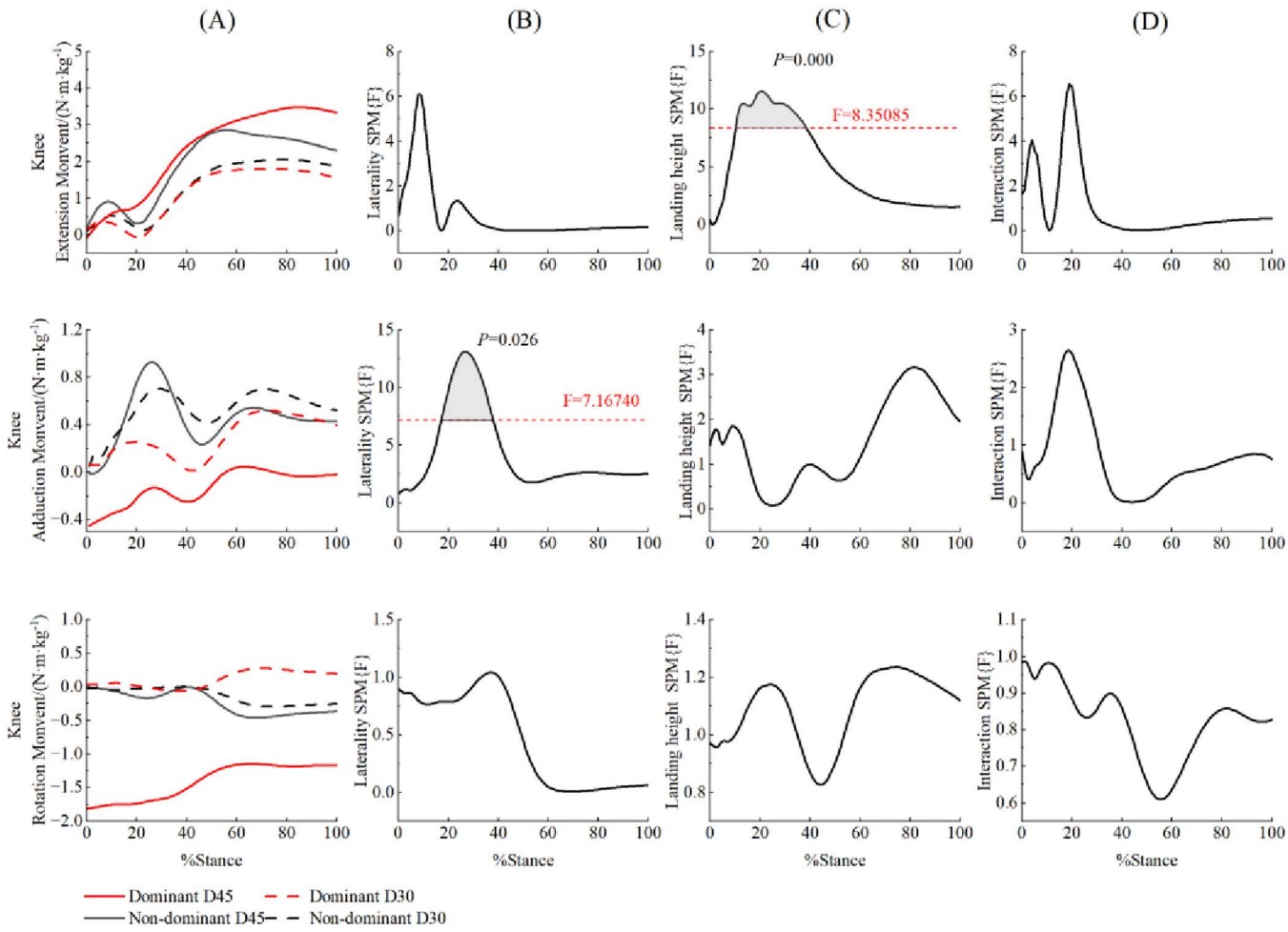

**Fig 7. Comparison of knee moments SPM{F} during the landing cushion phase for different side and drop heights.**

through coordinated muscular control and compensatory movement strategies [20,22]. During the landing cushioning phase, the non-dominant leg exhibits a more pronounced increase in abduction moment to compensate for uneven strength and stabilize overall posture. Increased hip abduction moment may shift the center of pressure laterally, leading to compensatory changes in adjacent joints and potentially increasing the risk of knee valgus and ACL injury [10,26–28]. The larger hip abduction moment in the non-dominant leg during the landing cushioning phase may impact the knee joint within the kinetic chain. Additionally, while SPM{F} analysis showed a significant increase in the non-dominant leg's hip abduction moment, no significant difference in external rotation moment was observed, likely due to the conservative significance threshold used in zero-dimensional statistical analyses to prevent Type I errors [18]. Previous research has shown that zero-dimensional models have limitations in deriving probabilistic inferences from one-dimensional dynamic data [15–17]. Although one-dimensional analysis may miss variations in knee joint external rotation, zero-dimensional models might still reveal significant differences. The SPM{F} results showed that the non-dominant ankle exhibited a higher internal rotation moment during landing. This finding may plausibly reflect an adaptive neuromuscular compensation strategy employed to preserve joint stability in response to asymmetric mechanical demands. Moreover, the non-dominant side, with relatively weaker muscle strength and less precise neural control, may require greater plantarflexion

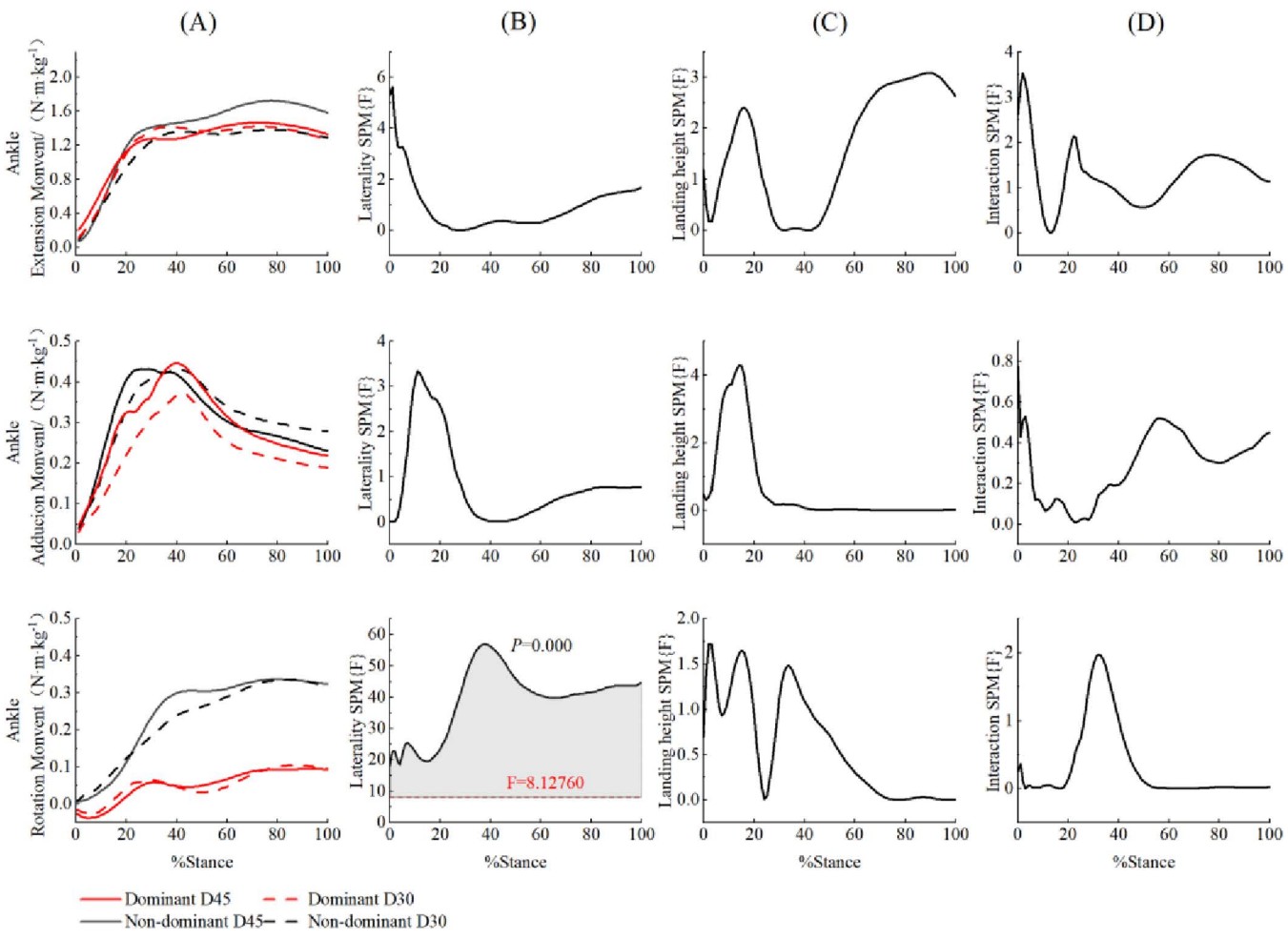

**Fig 8. Comparison of ankle moment SPM{F} during the landing cushioning phase for different side and drop heights.**

and internal rotation to maintain landing stability [9]. However, the extent to which joint morphology, motor control strategies, and limb dominance contribute to these asymmetries remains unclear and warrants further investigation through integrated biomechanical and neuromechanical analyses.

## Comparative analysis of different heights during the landing cushioning phase

Both two-way ANOVA and SPM{F} analyses confirmed that a 45 cm jump height generated significantly higher vertical GRF than a 30 cm jump. This aligns with physical principles: greater landing height increases vertical velocity with constant mass, requiring a higher impulse to counter momentum change. Due to the body's limited impact absorption capacity, excess force translates into higher GRF, amplifying landing impact. This finding supports Ali et al. [29], who reported a significant increase in knee flexion moment during landing from 45 cm compared to 30 cm. Previous studies have established that the knee joint, as the primary weight-bearing and energy-absorbing joint, plays a crucial role in impact force absorption during the cushioning phase [2,30]. Accordingly, our study showed that higher landing heights resulted in increased impact forces, placing greater load on the knee for absorption and cushioning [31]. This is consistent with Leppänen et al. [32], which reported a significant positive correlation between knee flexion moments and the risk of ACL

 

injury. Specifically, for every 10 N·m increase in knee flexion moment, the risk of ACL injury increased by 21% (HR = 1.21, 95% CI: 1.04–1.40, $p$ = 0.01) [2,20]. To reduce injury risk, rope jumping participants should consider controlling landing heights and gradually increasing them during training. Complementing this with strength and coordination exercises to enhance lower limb muscle and joint stability during high-intensity exercise.

## Limitations

This study, while providing valuable insights, has notable limitations. First, all participants were male, which limits the generalizability of the findings to female populations, who may exhibit different neuromuscular control strategies and joint loading patterns. Future studies should include female participants to explore potential sex-specific biomechanical responses during landing tasks. Second, although the CAST marker placement technique was used to minimize soft tissue artifacts, skin motion artifacts remain an inherent limitation in marker-based motion capture systems and may influence the accuracy of joint kinematic and kinetic measurements. Additionally, the study was conducted under a controlled laboratory setting, which does not fully replicate real-world rope skipping conditions. Moreover, the focus on lower limb cushioning during forward landings overlooks other landing strategies, and the role of upper limb and trunk dynamics remains unexplored. Future research should address these methodological limitations by incorporating more diverse populations, the influence of landing direction and the biomechanical contributions of the upper limbs and trunk, as these factors may significantly affect joint loading and injury risk [33].

## Conclusion

This study utilized Statistical Parametric Mapping to analyze landing biomechanics, identified greater hip abduction, knee external rotation, and ankle internal rotation moments on the non-dominant side, along with increased vertical ground reaction forces at higher landing heights, as key contributors to elevated joint loading. These findings suggest that neuromuscular coordination, postural control, and movement symmetry may be critical in mitigating injury risks during high-impact landing tasks. Therefore, future training strategies could consider improving motor control and functional strength, particularly in the non-dominant limb, to enhance stability and optimize landing performance. Regulating landing height and optimizing landing mechanics may further reduce injury risk. Future research should integrate these insights with in-depth biomechanical analyses to develop evidence-based training protocols and injury prevention strategies, ultimately improving both performance and safety in rope skipping.

## Supporting information

**S1 File.  zip Original data.**
(ZIP)

## Acknowledgments

The authors thank all participants for their valuable contributions and the Biomechanics Laboratory at Yanshan University for providing the experimental facilities that enabled this research.

## Author contributions

**Conceptualization:** Lian Duan, Yuan Gao, Feng Gao, Yang Sun, Naiyuan Tian.

**Formal analysis:** Lian Duan, Feng Gao.

**Investigation:** Lian Duan, Feng Gao, Naiyuan Tian.

**Methodology:** Lian Duan, Yuan Gao, Feng Gao, Yang Sun, Naiyuan Tian.

**Project administration:** Lian Duan, Feng Gao.

**Resources:** Yuan Gao, Yang Sun.

**Validation:** Yuan Gao, Yang Sun, Naiyuan Tian.

**Visualization:** Lian Duan, Feng Gao.

**Writing – original draft:** Lian Duan, Feng Gao.

**Writing – review & editing:** Lian Duan, Yuan Gao, Feng Gao, Yang Sun, Naiyuan Tian.

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
