## [Decision Letter · Decision Letter 0]

1 Apr 2025

PONE-D-25-09215Analysis of the Impact of Different Landing Heights and Lateralities on Lower Limb Joint Load Based on Statistical Parametric MappingPLOS ONE

Dear Dr. Gao,

Thank you for submitting your manuscript to PLOS ONE. After careful consideration, we feel that it has merit but does not fully meet PLOS ONE’s publication criteria as it currently stands. Therefore, we invite you to submit a revised version of the manuscript that addresses the points raised during the review process.

We look forward to receiving your revised manuscript.

Kind regards,

Seyed Hamed Mousavi

Academic Editor

PLOS ONE

Journal Requirements:

4. Please amend your authorship list in your manuscript file to include author Yuan Gao, Lian Duan , Feng Gao , Yang SUN , Naiyuan Tian .

5. Please remove your figures from within your manuscript file, leaving only the individual TIFF/EPS image files, uploaded separately. These will be automatically included in the reviewers’ PDF**.**

Reviewers' comments:

Reviewer's Responses to Questions

**Comments to the Author**

1. Is the manuscript technically sound, and do the data support the conclusions?

Reviewer #1: Yes

Reviewer #2: Partly

2. Has the statistical analysis been performed appropriately and rigorously? 

Reviewer #1: Yes

Reviewer #2: Yes

3. Have the authors made all data underlying the findings in their manuscript fully available?

Reviewer #1: Yes

Reviewer #2: Yes

4. Is the manuscript presented in an intelligible fashion and written in standard English?

Reviewer #1: Yes

Reviewer #2: Yes

5. Review Comments to the Author

Reviewer #1: This study seemed to have a purpose based on a gap in the literature. More vital information is needed in the method section. The results, discussion, and conclusions all related to the primary purpose of the study. One figure legend needs more information. Many word tense and word choice errors were identified and need correction. See pdf for specific comments.

Reviewer #2: 1.In the abstract, you use “knee's external abduction moment” (row 30), and “hip's external abduction moment” (row 31). There is no concept like “external abduction”. If you say that the moment is external, so how should we understand the “ankle joint internal rotation moment” (row 29)? Like internal moment or internal rotation in the ankle? Furthermore, you never mention them again in the text.

2.The purpose of the study is not clearly stated.

3.Your introduction section focuses entirely on the rope jumping process, while the study only includes landing from different heights, without a preceding flight phase. Indeed this investigation could relate to many other “landing” activities (for example all kinds of jumps). A revision in this section is needed. You perform the experimental investigation with rope-skipping enthusiasts, not professionals. How often do these enthusiasts practice rope-skipping, and do they practice other sports, leading to different leg strengths?

During standard rope-skipping man jump approximately 15-30 sm in height. How did you determine your 30 and 45 sm?

4.Please add a citation for “CAST lower limb model” and explain the abbreviation.

5.In addition to photos of the equipment, you can also add one or some participants during the experiment.

6.“This study analyzes biomechanical indicators …” (row 132). Which are they?

7.“The hip joint stabilizes the coronal plane by increasing abduction moments…” (row 229). This sentence structure is not correct. Hip joints do not stabilize any plane. Abduction movement is performed in that plane.

8.“Excessive hip abduction can shift the center of pressure outward, leading to compensatory changes in adjacent joints, increasing knee valgus and ACL injury risk ” (row 232). During drop landing, you do not observe “excessive hip abduction” which is over 50 degrees in the joint, but only increased abduction moment.

9.Rows 251-255. I recommend you 1) move it to the conclusion section; 2) do not be so confident in speculating the results with the need for “strength training ”. Joint stability depends on bone shape congruency, capsule and ligament elasticity, muscle strength, neuromuscular control, fatigue. Better find some previous literature sources, and think in the field of coordination, and motor control.

10.Some citations are missing from the text like 20, 22, 23…

6. PLOS authors have the option to publish the peer review history of their article (what does this mean? ). If published, this will include your full peer review and any attached files.

**Do you want your identity to be public for this peer review?** For information about this choice, including consent withdrawal, please see our Privacy Policy .

Reviewer #1: No

Reviewer #2: No

---

## [Author Response · Author response to Decision Letter 1]

8 Apr 2025

Response to Reviewer #1

Thank you for your valuable suggestions, here are the specific changes we have made

1.Errors in the text involving tenses and grammar have been corrected in accordance with your suggestions.

2.A new paragraph has been added to line 61 based on your suggestion.

3.Thank you for your valuable question regarding participant recruitment and quantity selection. In response, we have modified the original paragraph to read;

Study participants were recruited between 15 April 2024 and 23 April 2024 through campus bulletin board postings and social media platforms targeting university students at Yanshan University, Qinhuangdao, Hebei Province. The sample size was determined using G*Power software with the following parameters: effect size f = 0.4, α error probability = 0.05, power (1-β) = 0.8, number of groups = 1, number of measurements = 2, and correlation among repeated measures = 0.5. The minimum required sample size was calculated to be 15 participants [19]. To account for potential data loss due to movement failure, marker displacement, or invalid data, a 30% redundancy was added, Thus, 20 male rope-skipping enthusiasts participated in this study (age: 19.75 ± 0.86 years, height: 181.24 ± 4.62 cm, body mass�72.21± 7.66 kg, BMI: 22.06 ± 2.94 kg/m²). Data collection was completed during a 12-day experimental period following recruitment.

4.We have corrected the terminology to refer to "body mass" instead of "weight" to accurately reflect the use of kilograms (kg) as the unit of measurement.

5.In response to your question about the exclusion criteria for subjects, we have added a detailed description of the inclusion and exclusion criteria in Section Participants. which reads:

All participants were recruited based on the following inclusion criteria: (1) no congenital lower extremity deformities or acquired musculoskeletal disorders; (2) no history of lower extremity joint injuries within the past 6 months; and (3) sufficient physical ability to perform the task of stair walking. Exclusion criteria included (1) a history of lower extremity surgery; (2) use of medications affecting neuromuscular function within the past 12 months; (3) the presence of neuromuscular, psychologic, or cardiovascular disorders; and (4) the presence of acute pain symptoms during the test.

6.In response to your question about subjects familiarizing themselves with the movements and warming up. We have added content notes to the “Single Leg Landing Test” section as follows:

Prior to the formal test, participants underwent a 6-minute familiarization process, including a movement demonstration and practice trials. Additionally, a 10-minute warm-up was conducted, consisting of 5 minutes of jogging and dynamic stretching, followed by 5 minutes of light jumping and landing exercises to prepare participants for the task.

7.Regarding your question: who placed the markers? what was their level of experience? We have added a description of the relevant content in the Data collection section, as follows:

All markers were applied by the same experienced and professionally trained researcher, following standardized procedures and anatomical landmark definitions to ensure data accuracy and consistency.

8.In response to your suggestion, added references to CAST lower extremity modeling.

[23] Cappozzo, A., Catani, F., Della Croce, U., & Leardini, A. (1995). Position and orientation in space of bones during movement: anatomical frame definition and determination. Clinical biomechanics, 10(4), 171-178. https://doi.org/10.1016/0268-0033(95)91394-T.

9.Regarding your insightful question about data filtering.

In this study, a fourth-order Butterworth low-pass filter with a cutoff frequency of 12 Hz was mainly applied to smooth the trajectory data. The joint torque data were not filtered because they were derived using inverse dynamics, where the input data (including motion trajectory and filtered ground reaction force (GRF) data) were processed before calculation. Although the raw GRF data were collected using a high-precision Kistler force plate, the same 12 Hz low-pass filter was applied to the GRF data to ensure consistency with the frequency characteristics of the motion data, so the torque data itself did not need secondary filtering.

10. Regarding your question about the statement of Joint moments.

The moments mentioned in this study refer to joint moments, which are calculated by inverse dynamics methods, not internal net moments. Joint moments are calculated based on kinematic trajectories and ground reaction force (GRF) data.

11.Regarding the incorrect structure of the first sentence of the result section, I have modified it. The specific content is as follows:

Two-way ANOVA and SPM{F} analyses were performed to examine the effects of laterality (dominant vs non-dominant) and landing height (30 cm vs 45 cm) on lower extremity joint loads. The results showed that there was no significant interaction between laterality and height (P > 0.05). This suggests that these two factors independently affect joint loads and have no combined effect.

12.Added the p-values in Table 2 based on your suggestion, and added explanation for the peak moment in the table.

13.According to your suggestion, the problem to be solved and the purpose of the study are explained before the discussion begins. The specific contents is as follows:

Previous studies have primarily focused on single-limb mechanics, assuming bilateral symmetry, which limits understanding of limb asymmetries in dynamic activities. Given the high incidence of lower limb injuries, examining these differences is crucial. This study aimed to investigate the effects of landing height and limb lateralities on joint loading, addressing this gap and providing insights for injury prevention and performance optimization.

14.Regarding the description of the positive correlation in the cited Leppänen article, it has been revised in the text as follows:

This is consistent with Leppänen et al. [32], which reported a significant positive correlation between knee flexion moments and the risk of ACL injury. Specifically, for every 10 N·m increase in knee flexion moment, the risk of ACL injury increased by 21% (HR = 1.21, 95% CI: 1.04–1.40, p = 0.01) [2, 20].

Because Leppänen et al. reported hazard ratios (HR) rather than correlation coefficients (r). Due to methodological differences, the assessment of ACL injury risk is typically conducted using survival analysis and regression models; therefore, r values were not provided in their study.

Response to Reviewer #2

1.Reviewer #2: Suggestion: In the abstract, you use “knee's external abduction moment” (row 30), and “hip's external abduction moment” (row 31). There is no concept like “external abduction”. If you say that the moment is external, so how should we understand the “ankle joint internal rotation moment” (row 29)? Like internal moment or internal rotation in the ankle? Furthermore, you never mention them again in the text.

Author: Thank you for pointing out the problem of inconsistent and ambiguous terminology describing joint moments in the abstract. We agree that phrases such as “external abduction moment” are imprecise and may cause confusion. To clarify, we have revised the wording as follows: “The knee's abduction moment significantly increased during the 17%–37% landing phase, and the hip's abduction moment significantly increased during the 86%–100% phase.”

2.Reviewer #2: The purpose of the study is not clearly stated.

Author: Thank you for your helpful suggestions on clarifying the purpose of the study. In response, we have revised the opening sentence of the Abstract to explicitly state the aim of the study. In addition, we have updated the final paragraph of the Introduction to clearly articulate the specific objectives, research focus, and intended contributions of the work.

3.Reviewer #2: Your introduction section focuses entirely on the rope jumping process, while the study only includes landing from different heights, without a preceding flight phase. Indeed this investigation could relate to many other “landing” activities (for example all kinds of jumps). A revision in this section is needed. You perform the experimental investigation with rope-skipping enthusiasts, not professionals. How often do these enthusiasts practice rope-skipping, and do they practice other sports, leading to different leg strengths?

Author: Thank you for your insightful comments regarding the focus of the introduction and the sample population studied. Regarding the focus on rope skipping, we acknowledge that the introduction may have emphasized this specific activity. However, the primary goal of the study was to explore landing mechanics, which are not limited to rope skipping but extend to various dynamic landing activities such as jumps from different heights. The focus on rope skipping enthusiasts was driven by the practical need to assess biomechanics in a commonly practiced activity where landing injuries are prevalent, thus providing relevant data for injury prevention and performance optimization in this context.

In terms of the sample population, we understand the importance of leg strength and the potential impact of training frequency on biomechanics. However, the participants in this study were specifically selected to represent rope skipping enthusiasts with regular training, and we did not explore additional factors such as cross-training or other sports. While the potential variability in leg strength could be considered in future research, the current study's aim was to focus on the biomechanical effects of landing height and limb lateralities, which are independent of the participants’ leg strength or training history. Future studies could expand this research by incorporating athletes with different training backgrounds or exploring variations in landing techniques across a broader range of activities.

Reviewer #2: During standard rope-skipping man jump approximately 15-30 sm in height. How did you determine your 30 and 45 sm?

Author: Thank you for your valuable comments regarding the selection of 30 cm and 45 cm as landing heights. In standard rope skipping, the vertical displacement during flight is relatively small (typically 15–30 cm), as noted. However, the heights of 30 cm and 45 cm were intentionally selected to simulate increased landing intensity conditions, which are commonly encountered in freestyle rope skipping or other jumping-related activities where athletes perform complex aerial movements before landing.

These controlled drop heights are widely used in biomechanical research to safely reproduce variable landing loads and assess their effects on joint kinetics and kinematics under standardized conditions. The use of 30 cm and 45 cm also aligns with prior studies analyzing single-leg or drop landings to provoke meaningful biomechanical responses, particularly in joint loading and ground reaction forces. Therefore, while these heights may exceed typical skipping jump heights, they provide a practical and validated model for investigating the effects of increased landing demand, which is relevant not only to rope skipping but to various athletic movements involving unilateral landings.

4.Reviewer #2: Please add a citation for “CAST lower limb model” and explain the abbreviation.

Author: Thank you for pointing out this important problemand we have made the following changes.

During trials, participants wore study-specific tights, and after warming up, 20 infrared reflective markers were placed on anatomical landmarks according to the CAST (Calibrated Anatomical Systems Technique) lower limb model, which employs static calibration and marker tracking to establish joint coordinate systems while minimizing soft tissue artifacts [23].

[23]Cappozzo, A., Catani, F., Della Croce, U., & Leardini, A. (1995). Position and orientation in space of bones during movement: anatomical frame definition and determination. Clinical biomechanics, 10(4), 171-178. https://doi.org/10.1016/0268-0033(95)91394-T.

5.Reviewer #2: In addition to photos of the equipment, you can also add one or some participants during the experiment.

Author: Thank you for your suggestion, to which we added images of the participants.

Fig. 3. Participants performing the single-leg landing task from two different heights (30 cm and 45 cm).

Note: Participants stood on the jumping box with one leg lifted, then leaned forward and landed on the test leg while keeping their hands on their hips. The two heights correspond to the experimental conditions used for data collection.

6.Reviewer #2: “This study analyzes biomechanical indicators …” (row 132). Which are they?

Author: We appreciate your attention to detail. The biomechanical indicators analyzed in this study include vertical ground reaction force (vGRF) and three-dimensional joint moments of the hip, knee, and ankle during the landing cushioning phase. Specifically, the joint moments include:

Hip: flexion/extension, abduction/adduction, internal/external rotation

Knee: flexion/extension, varus/valgus, internal/external rotation

Ankle: plantarflexion/dorsiflexion, inversion/eversion, internal/external rotation

These indicators are described in detail in Section Statistical Analysis and summarized throughout the Results section (Tables 1–2 and Figures 4–7).

7.Reviewer #2: “The hip joint stabilizes the coronal plane by increasing abduction moments…” (row 229). This sentence structure is not correct. Hip joints do not stabilize any plane. Abduction movement is performed in that plane.

Author: Thank you for your insightful comments! We agree that the hip joint itself does not stabilize any anatomical plane. Rather, we intended to express that hip abduction torque, which occurs in the frontal plane, contributes to dynamic postural control during landing. We have revised the sentence accordingly to avoid misinterpretation.

During landing, increased hip abduction moment in the frontal plane contributes to maintaining postural stability through coordinated muscular control and compensatory movement strategies.

8.Reviewer #2: “Excessive hip abduction can shift the center of pressure outward, leading to compensatory changes in adjacent joints, increasing knee valgus and ACL injury risk ” (row 232). During drop landing, you do not observe “excessive hip abduction” which is over 50 degrees in the joint, but only increased abduction moment.

Author: Thank you for pointing this out. We agree that “excessive hip abduction” may be misleading in the context of our findings, as our study did not assess hip abduction angles but rather observed increased hip abduction moments on the non-dominant side during landing. We have revised the sentence to accurately reflect this and avoid confusion between joint angle and joint moment.

The revised paragraph reads: Increased hip abduction moment may shift the center of pressure laterally, leading to compensatory changes in adjacent joints and potentially increasing the risk of knee valgus and ACL injury.

9.Reviewer #2: Rows 251-255. I recommend you 1) move it to the conclusion section; 2) do not be so confident in speculating the results with the need for “strength training ”. Joint stability depends on bone shape congruency, capsule and ligament elasticity, muscle strength, neuromuscular control, fatigue. Better find some previous literature sources, and think in the field of coordination, and motor control.

Author: Thank you for the valuable feedback. We agree that joint stability is a multifactorial phenomenon influenced by not only muscle strength but also ligament properties, joint congruency, neuromuscular control, and fatigue. In response, the statement has been moved to the conclusion section as recommended.

10.Reviewer #2: Some citations are missing from the text like 20, 22, 23…

Thank you for pointing out the problem regarding missing citations. Upon careful review, we would like to clarify that reference 20 is cited in Section Participants of the manuscript. Reference 22 is included in Section Single-leg landing test. References 23 and 24 are cited in the Data collection and Data proces

---

## [Decision Letter · Decision Letter 1]

17 Apr 2025

PONE-D-25-09215R1Analysis of the Impact of Different Landing Heights and Lateralities on Lower Limb Joint Load Based on Statistical Parametric MappingPLOS ONE

Dear Dr. Gao,

Thank you for submitting your manuscript to PLOS ONE. After careful consideration, we feel that it has merit but does not fully meet PLOS ONE’s publication criteria as it currently stands. Therefore, we invite you to submit a revised version of the manuscript that addresses the points raised during the review process.

We look forward to receiving your revised manuscript.

Kind regards,

Seyed Hamed Mousavi

Academic Editor

PLOS ONE

Journal Requirements:

Reviewers' comments:

Reviewer's Responses to Questions

**Comments to the Author**

1. If the authors have adequately addressed your comments raised in a previous round of review and you feel that this manuscript is now acceptable for publication, you may indicate that here to bypass the “Comments to the Author” section, enter your conflict of interest statement in the “Confidential to Editor” section, and submit your "Accept" recommendation.

Reviewer #1: All comments have been addressed

2. Is the manuscript technically sound, and do the data support the conclusions?

Reviewer #1: Yes

3. Has the statistical analysis been performed appropriately and rigorously? 

Reviewer #1: Yes

4. Have the authors made all data underlying the findings in their manuscript fully available?

Reviewer #1: Yes

5. Is the manuscript presented in an intelligible fashion and written in standard English?

Reviewer #1: No

6. Review Comments to the Author

Reviewer #1: Major issues have been addressed. However, some additional information is needed yet in the methods section, there remain several word tense and word choice errors, and a couple of comments related to your discussion need to be considered. See specific comments on the revised (red highlighted) manuscript.

7. PLOS authors have the option to publish the peer review history of their article (what does this mean? ). If published, this will include your full peer review and any attached files.

**Do you want your identity to be public for this peer review?** For information about this choice, including consent withdrawal, please see our Privacy Policy .

Reviewer #1: No

---

## [Author Response · Author response to Decision Letter 2]

21 Apr 2025

Response to Reviewer

Thank you for your valuable suggestions, here are the specific changes we have made

1.Errors in the text involving tenses and grammar have been corrected in accordance with your suggestions.

2.A fourth-order Butterworth low-pass filter with a cutoff frequency of 12 Hz was applied to smooth the data, as described by Kristianslund et al. [24], Lenz et al. [25] and Monfort-Torres et al. [21].

Thank you for your valuable questions regarding the data smoothing process in this study. In response, we have modified the original paragraph as follows

A fourth-order Butterworth low-pass filter with a cutoff frequency of 12 Hz was applied to smooth the trajectory data, as described by Kristianslund et al. [24], Lenz et al. [25] and Monfort-Torres et al. [21].

3.After reviewing the relevant literature, we have changed “joint moments” to “Net internal joint moments” according to your suggestion, thank you for pointing out the key issues.

4.During landing, increased[20] hip abduction moment in the frontal plane contributes to maintaining postural stability through coordinated muscular control and compensatory movement strategies[20, 22].

The original text has been modified in accordance with your suggestions: During landing, increased hip abduction moment contributes to maintaining postural stability through coordinated muscular control and compensatory movement strategies [20, 22].

5.This is likely due to the ankle's complex anatomy, which is inherently more prone to deformation upon impact.

Thank you for your suggestion to point out the content of this paragraph, we have revised the original content according to your suggestion:

This finding may plausibly reflect an adaptive neuromuscular compensation strategy employed to preserve joint stability in response to asymmetric mechanical demands. Moreover, the non-dominant side, with relatively weaker muscle strength and less precise neural control, may require greater plantarflexion and internal rotation to maintain landing stability [9]. However, the extent to which joint morphology, motor control strategies, and limb dominance contribute to these asymmetries remains unclear and warrants further investigation through integrated biomechanical and neuromechanical analyses.

6.This finding supports Ali et al. [29], revealing a significant increase in knee flexion moment during landing from 45 cm compared to 30 cm.

Thank you for your suggestion to point out the content of this sentence, we have modified the original content according to your suggestion:

This finding supports Ali et al. [3], who reported a significant increase in knee flexion moment during landing from 45 cm compared to 30 cm.

7.To reduce injury risk, rope jumping participants[20]should control landing heights and gradually increase them during training.

Thank you for your suggestion to point out the content of this sentence, we have modified the original content according to your suggestion:

To reduce injury risk, rope jumping participants should consider controlling landing heights and gradually increasing them during training.

8.Thank you for your suggestions on the limitations section of this study, we have revised the original paragraph as follows:

This study, while providing valuable insights, has notable limitations. First, all participants were male, which limits the generalizability of the findings to female populations, who may exhibit different neuromuscular control strategies and joint loading patterns. Future studies should include female participants to explore potential sex-specific biomechanical responses during landing tasks. Second, although the CAST marker placement technique was used to minimize soft tissue artifacts, skin motion artifacts remain an inherent limitation in marker-based motion capture systems and may influence the accuracy of joint kinematic and kinetic measurements. Additionally, the study was conducted under a controlled laboratory setting, which does not fully replicate real-world rope skipping conditions. Moreover, the focus on lower limb cushioning during forward landings overlooks other landing strategies, and the role of upper limb and trunk dynamics remains unexplored. Future research should address these methodological limitations by incorporating more diverse populations, the influence of landing direction and the biomechanical contributions of the upper limbs and trunk, as these factors may significantly affect joint loading and injury risk [33].

9.This study, based on Statistical Parametric Mapping, identified greater hip abduction�…

Thank you for your suggestions regarding the conclusion section of this study, we have modified the original sentence as follows:

This study utilized Statistical Parametric Mapping to analyze landing biomechanics, …

Thanks again for your valuable suggestions.

---

## [Decision Letter · Decision Letter 2]

29 Apr 2025

Analysis of the Impact of Different Landing Heights and Lateralities on Lower Limb Joint Load Based on Statistical Parametric Mapping

PONE-D-25-09215R2

Dear Dr. Gao,

We’re pleased to inform you that your manuscript has been judged scientifically suitable for publication and will be formally accepted for publication once it meets all outstanding technical requirements.

Kind regards,

Seyed Hamed Mousavi

Academic Editor

PLOS ONE

Additional Editor Comments (optional):

Reviewers' comments:

Reviewer's Responses to Questions

**Comments to the Author**

1. If the authors have adequately addressed your comments raised in a previous round of review and you feel that this manuscript is now acceptable for publication, you may indicate that here to bypass the “Comments to the Author” section, enter your conflict of interest statement in the “Confidential to Editor” section, and submit your "Accept" recommendation.

Reviewer #1: All comments have been addressed

2. Is the manuscript technically sound, and do the data support the conclusions?

Reviewer #1: Yes

3. Has the statistical analysis been performed appropriately and rigorously? 

Reviewer #1: Yes

4. Have the authors made all data underlying the findings in their manuscript fully available?

Reviewer #1: Yes

5. Is the manuscript presented in an intelligible fashion and written in standard English?

Reviewer #1: Yes

6. Review Comments to the Author

Reviewer #1: Thank you for addressing previous and subsequent comments/questions/suggestions. This most recent revision has addressed the final concerns.

7. PLOS authors have the option to publish the peer review history of their article (what does this mean? ). If published, this will include your full peer review and any attached files.

**Do you want your identity to be public for this peer review?** For information about this choice, including consent withdrawal, please see our Privacy Policy .

Reviewer #1: No

---

## [Editor Report · Acceptance letter]

PONE-D-25-09215R2

PLOS ONE

Dear Dr. Gao,

I'm pleased to inform you that your manuscript has been deemed suitable for publication in PLOS ONE. Congratulations! Your manuscript is now being handed over to our production team.

Kind regards,

on behalf of

Dr. Seyed Hamed Mousavi

Academic Editor

PLOS ONE